# Germline Variants in 32 Cancer-Related Genes among 700 Chinese Breast Cancer Patients by Next-Generation Sequencing: A Clinic-Based, Observational Study

**DOI:** 10.3390/ijms231911266

**Published:** 2022-09-24

**Authors:** Liu Yang, Fei Xie, Chang Liu, Jin Zhao, Taobo Hu, Jinbo Wu, Xiaotao Zhao, Shu Wang

**Affiliations:** 1Department of Breast Disease Center, Peking University People’s Hospital, Beijing 100044, China; 2Department of Clinical Laboratory, Peking University People’s Hospital, Beijing 100044, China

**Keywords:** breast cancer, next-generation sequencing, Chinese population, germline mutations, BRCA1/2

## Abstract

Breast cancer (BC) is associated with hereditary components, and some deleterious germline variants have been regarded as effective therapeutic targets. We conducted a clinic-based, observational study to better understand the distribution of deleterious germline variants and assess any clinicopathological predictors related to the variants among Chinese BC patients using a 32 cancer-related genes next-generation sequencing panel. Between November 2020 and February 2022, a total of 700 BC patients were recruited, and 13.1% (92/700) of them carried deleterious germline variants in 15 cancer-related genes, including 37 (37/700, 5.3%) in *BRCA2*, 29 (29/700, 4.1%) in *BRCA1*, 8 (8/700, 1.1%) in *PALB2*, 4 (4/700, 0.6%) in *NBN*, 3 (3/700, 0.4%) in *MRE11A*, 3 (3/700, 0.4%) in *TP53* and 12 (12/700, 1.7%) in other genes. There were 28 novel variants detected: 5 in *BRCA1*, 14 in *BRCA2*, and 9 in non-*BRCA1/2* genes. The variants in panel genes, HRR (homologous recombination repair)-related genes, and *BRCA1/2* were significantly associated with the following clinicopathological factors: age at the initial diagnosis of BC, family history of any cancer, molecular subtype, Ki-67 index, and hereditary risk. In conclusion, we further expanded the spectrum of germline deleterious variants in Chinese BC patients, and the clinicopathological predictors of variants were identified to facilitate clinical genetic testing and counseling for appropriate individuals.

## 1. Introduction

Breast cancer (BC) is the most commonly diagnosed malignancy worldwide [1]. BC is associated with hereditary components and approximately 10% of unselected cases are reported to carry a pathogenic germline variant in cancer-related genes [2,3]. For BC patients at hereditary high risk, even about 24% of them carried detrimental germline variants [4].

So far, germline variants in about 10 susceptibility genes have been confirmed to be involved in tumorigenesis and increase the BC risk, mainly including genes involved in DNA repair (*BRCA1*, *BRCA2*, *ATM*, *CHEK2*, *PALB2*, *BARD1*, *RAD51C*, and *RAD51D*) and genes caused pleiotropic tumor syndromes (*TP53*, *CDH1*, *PTEN*, *STK11*, and *NF1*) [5,6]. The current clinical trials have selected some significant germline variants which can be regarded as effective therapeutic targets [7,8,9]. For example, germline *BRCA1/2* variants were reported to synergize with Poly (ADP-ribose) polymerase (PARP) inhibitors to inhibit tumor growth and improve survival in patients with early or metastatic BC [7,8]. Therefore, it is essential to screen for breast cancer susceptibility genes, and the National Comprehensive Cancer Network (NCCN) guidelines recommend multi-gene testing and genetic counseling in appropriate individuals [10].

In recent years, next-generation sequencing (NGS) has provided an efficient and cost-effective platform to sequence multiple genes simultaneously, which were not detected in a single gene test [11]. It has become a routine clinical practice in many Western countries that perform phenotypically directed multi-gene panel screening to assess for pathogenic changes in multiple genes. Screening for variants in high-penetrance predisposition genes is a consensus, especially *BRCA1/2* [10]. However, testing for other cancer-related genes, including low to moderate-penetrance predisposition genes, remains controversial due to insufficient evidence for the prevalence of variants and increased risk. Furthermore, predictive factors for germline variants have not been identified, therefore, leading to a limitation of clinical utility.

In the previous study, we identified four novel germline *BRCA* variants from 78 Chinese patients with BC by the NGS and conducted functional assays to understand the pathogenic mechanism of these variants [12]. However, the significance was limited due to the small sample size. Thus, we conducted a large clinic-based cohort of Chinese BC patients to further explore the spectrum of germline variants in 32 cancer-related genes. We aimed to better understand the distribution of the deleterious germline variants and assess any clinicopathological predictors associated with the germline variants.

## 2. Results

### 2.1. Patients Characteristics

A total of 700 patients diagnosed with BC were recruited, including 695 females and 5 males. Chinese Han patients accounted for 99.4% (696/700) of all patients included, and ethnic minority patients accounted for 0.6% (6/700). The median age at initial diagnosis of BC among overall patients was 50 years, ranging from 23 to 87 years. The clinicopathological characteristics of 700 BC patients included in this study are listed in Table 1.

### 2.2. Germline Variant Spectrum

Among all patients, 13.1% (92/700) carried deleterious germline variants in 15 cancer-related genes. Except for three (3/700, 0.4%) patients with *TP53* variants, 12.7% (89/700) patients carried homologous recombination repair (HRR)-related variants followed by *BRCA2* (37/700, 5.3%), *BRCA1* (29/700, 4.1%), *PALB2* (8/700, 1.1%), *NBN* (4/700, 0.6%), *MRE11A* (3/700, 0.4%), *BARD1* (2/700, 0.3%), *CHEK1* (2/700, 0.3%), *CHEK2* (2/700, 0.3%), *ATR* (1/700, 0.1%), *BRIP1* (1/700, 0.1%), *FANCL* (1/700, 0.1%), *RAD51C* (1/700, 0.1%), *RAD51D* (1/700, 0.1%), and *RAD54L* (1/700, 0.1%). There were four patients carried two distinct variants: *BRCA1* with *MRE11A*, *BRCA2* with *NBN*, *BRCA2* with *FANCL*, and *ATR* with *CHEK1*. Protein truncating, including frameshift (41/96, 42.7%) and nonsense (33/96, 34.4%), were common types of variants. The spectrum of deleterious germline variants in 32 cancer-related genes among 700 BC patients is shown in Figure 1.

A total of 25 *BRCA1* deleterious variants were detected in 29 patients, including 25 (25/29, 86.2%) protein-truncating (15 frameshift and 10 nonsense), 2 (2/29, 6.9%) missense, 1 (1/29, 3.4%) start lost, and 1 (1/29, 3.4%) E15 deletion. The most frequent variant in *BRCA1* was c.5470_5477del (p.I1824Dfs*3, *n* = 3), which was considered a founder variant in the Chinese Han patients with BC or ovarian cancer [13,14]. Other recurrent *BRCA1* variants were c.2998G > T (p.E1000*, *n* = 2) and c.228_229delinsAAAGTG (p.S76Rfs*6, *n* = 2). Five novel variants in *BRCA1* were firstly detected: c.228_229delinsAAAGTG (p.S76Rfs*6, *n* = 2), c.2998G > T (p.E1000*, *n* = 2), c.242_244delinsT (p.Q81Lfs*3, *n* = 1), c.4162del (p.Q1388Rfs*5, *n* = 1), and c.5090delG (p.C1697Lfs*5, *n* = 1). Locations of variants in the BRCA1 proteins are presented in Figure 2a, and 13 variants detected in 15 (15/29, 51.7%) patients were in BRCA1 protein domains: Two in the RING finger domain, three in the nuclear localization signal (NLS) domain, four in ethylene insensitive 3 domain, and four in BRCA carboxyl-terminus (BRCT) domain. The variants occurred most frequently in the BRCT domain with a total of six (20.7%, 6/29) variants, three of which were p.I1824Dfs*3.

*BRCA2* variants were the most frequent in patients included in the study. A total of 35 deleterious variants were detected in 37 patients, including 31 (31/37, 83.8%) protein-truncating (18 frameshift and 13 nonsense), 3 (3/37, 8.1%) splice site variants, 1 (1/37, 2.7%) missense, 1 synonymous (1/37, 2.7%), and 1 (1/37, 2.7%) E22-E24 deletion. The frequent *BRCA2* variant c.2808_2811del (p.A938Pfs*21) in non-Ashkenazi BC patients [15] was screened in two Chinese BC patients. Another recurrent *BRCA2* variant was c.9382C > T (p.R3128*, *n* = 2). Fourteen (14/35, 40%) novel *BRCA2* variants were observed firstly: c.1843_1849del7 (p.N615Qfs*27, *n* = 1), c.2636dup (p.E880*, *n* = 1), c.4914_4915insA (p.V1639Sfs*3, *n* = 1), c.5367del (p.V1790*, *n* = 1), c.5851dup (p.S1951Kfs*9, *n* = 1), c.6004G > T (p.E2002*, *n* = 1), c.6230delA (p.K2077Rfs*4, *n* = 1), c.6450_6451insA (p.V2151Sfs*25, *n* = 1), c.6938-32dup (*n* = 1), c.7163delC (p.T2388Kfs*6, *n* = 1), c.7516-3T > C (M2235Ffs*5, *n* = 1), c.7690_7691del (p.T2564*, *n* = 1), c.7984del (p.T2662Rfs*11, *n* = 1), and c.8800C > T (p.Q2934*, *n* = 1). Locations of variants in the BRCA2 proteins are presented in Figure 2b. Among them, 73.0% (27/37) occurred in non-domain loci, and only 27.0% (10/37) were detected in BRCA2 protein domain loci: Five in the BRC repeats domain, two in helix domain, two in the NLS domain (both are p.R3128*), and one in oligonucleotide binding-folds (OB folds) domain.

Among 30 (30/700, 4.3%) patients carried with 26 non-*BRCA* variants, *PALB2* variants (8/700, 1.1%) were the most frequent with seven protein-truncating (four frameshift and three nonsense) and one splice site variant. One recurrent variant (c.1912T > C (p.S638P)) was observed in all four (4/700, 0.6%) patients with *NBN* variant. Three (3/700, 0.4%) patients carrying *MRE11A* variants: One recurrent variant (c.1897C > T (p.R633*), *n* = 2) and one missense. A total of nine (9/26, 34.6%) novel variants in non-*BRCA* genes was firstly detected: *PALB2* c.23_30del (p.P8Lfs*2, *n* = 1), *PALB2* c.2605del (p.S869Pfs*2, *n* = 1), *MRE11A* c.1985C > T (p.T662I, *n* = 1), *CHEK1* c.760_773del14 (p.I254Hfs*5, *n* = 1), *CHEK1* c.613+2T > C (*n* = 1), *CHEK2* c.98C > G (p.S33*, *n* = 1), *ATR* c.4246_4247insT (p.S1416Ffs*14, *n* = 1), *BRIP1* c.1781T > G (p.L594*, *n* = 1), and *RAD54L* c.537_547 (p.D183Afs*9, *n* = 1). Locations of variants in the non-BRCA proteins are presented in Appendix A.

### 2.3. Association between Deleterious Germline Variants and Clinicopathological Characteristics

#### 2.3.1. Association between Deleterious Germline Variants and Age at the Initial Diagnosis of BC

The variant rates of panel-gene (*p* < 0.001) and *BRCA1/2* (*p* < 0.001) decreased with age at the initial diagnosis of BC (shown in Figure 3a). The distribution of variants according to age at the initial diagnosis of BC is shown in Figure 3b. For patients aged 45 years or less (early-onset breast cancer patients, *n* = 288), a total of 51 variants were detected in 50 (50/288, 17.4%) patients, followed by *BRCA2* (45.1%, 23/51), *BRCA1* (35.3%, 18/51), *PALB2* (7.8%, 4/51), *TP53* (5.9%, 3/51), and other genes (5.9%, 3/51). For patients older than 45 years (*n* = 412), the variant rate was only 10.2% (42/412) with 45 variants detected, including *BRCA2* (31.1%, 14/45), *BRCA1* (24.4%, 11/45), *PALB2* (8.9%, 4/45), *NBN* (8.9%, 4/45), *BARD1* (4.4%, 2/45), and other genes (22.2%, 10/45). *BRCA1* and *BRCA2* variants were enriched in early-onset patients (*BRCA1*: 6.7% vs. 2.6%, *p* = 0.008; *BRCA2*: 8.5% vs. 3.3%, *p* = 0.02). *TP53* variants were only detected in early-onset patients, and *BARD1* variants only in patients older than 55 years.

#### 2.3.2. Association between Deleterious Germline Variants and Family History of Cancer

Of 700 patients, 30.4% (213/700) had a family history of any cancer. Among them, 59.6% (127/213) presented a positive family history of *BRCA*-related cancer (breast cancer, *n* = 112; epithelial ovarian cancer, *n* = 6; exocrine pancreatic cancer, *n* = 12; prostate cancer, *n* = 4), 23.5% (50/213) had a positive family history of gastrointestinal cancer, and 19.2% (41/213) presented a positive family history of lung cancer (shown in Appendix A).

The variant rates of panel-gene (17.8% vs. 11.1%, *p* = 0.021) and *BRCA1/2* (14.1% vs. 7.4%, *p* = 0.008) were higher in patients who presented a positive family history of any cancer (shown in Figure 4a). However, there was no significant association between the deleterious variants and the types of family history of cancer (shown in Appendix A). The distribution of variants according to the family history of any cancer is shown in Figure 4b. Among patients presenting a positive family history of any cancer (*n* = 213), a total of 39 deleterious variants were detected in 38 patients, including *BRCA1* (41.0%, 16/39), *BRCA2* (35.9%, 14/39), *PALB2* (5.1%, 2/39), *TP53* (5.1%, 2/39), and other genes (12.8%, 5/39). Of patients who did not have a positive family history of any cancer (*n* = 487), 57 deleterious variants were detected in 54 patients, including *BRCA2* (40.4%, 23/57), *BRCA1* (22.8%, 13/57), *PALB2* (10.5%, 6/57), *NBN* (5.3%, 3/57), *MRE11A* (5.3%, 3/57) and other genes (15.8%, 9/57). *BRCA1* variants were more frequent in patients presenting a positive family history of any cancer compared with the patients who did not have a positive family history of any cancer (7.5% vs. 2.7%, *p* = 0.003).

#### 2.3.3. Association between Deleterious Germline Variants and Molecular Subtype of BC

Of 700 patients, 57.0% (399/700) were hormone receptor (HR)-positive/human epidermal growth factor receptor 2 (Her2)-negative, 23.4% (164/700) were HR-negative/Her2-negative (triple-negative, TN), 10% (70/700) were HR-positive/Her2-positive, and 9.6% (67/700) were HR-negative/Her2-positive. Both panel-gene (*p* = 0.001) and *BRCA1/2* (*p* < 0.001) variants were associated with the molecular subtype of BC (shown in Figure 5a). The distribution of variants according to the molecular subtype of BC is shown in Figure 5b. TNBC patients had the highest frequency of panel-gene variants with a rate of 22.0% (36/164), and a total of 37 variants were detected among them, including *BRCA1* (51.4%, 19/37), *BRCA2* (24.3%, 9/37), *BARD1* (5.4%, 2/37), *PALB2* (5.4%, 2/37), and other genes (13.5%, 5/37). The panel-gene variant rate of HR-positive (any Her2 expression status) BC patients was 11.1% (52/469), and a total of 54 variants were detected in these patients, including *BRCA2* (50%, 27/54), *BRCA1* (18.5%, 10/54), *PALB2* (11.1%, 6/54), *NBN* (5.6%, 3/54), and other genes (14.8%, 8/54). Her2-positive (any HR expression status) BC patients presented the lowest variant rate of panel-gene (7.3%, 10/137), and 11 variants were detected among them, mainly including *BRCA2* (45.5%, 5/11) and *TP53* (18.2%, 2/11).

*BRCA1* variants were associated with TN phenotype (11.6% vs. 1.9%, *p* < 0.001) and *BRCA2* variants were enriched in patients with an HR-positive BC (5.8% vs. 4.3%, *p* = 0.427). As for non-*BRCA* genes, *PALB2* (1.5% vs. 0.9%, *p* = 0.725) and *CHEK2* (0.4% vs. 0.0%) variants were commonly detected in HR-positive BC patients. *TP53* variants were more in patients with an HR-negative/Her2-positive BC (1.2% vs. 0.2%, *p* = 0.139). *BARD1* variants were all detected in TNBC patients (1.2% vs. 0.0%).

In addition, the deleterious germline variants were associated with the Ki-67 index. Patients with a high expression (Ki-67 index > 20%) of Ki-67 (*n* = 326) had a higher prevalence of panel-gene (18.7% vs. 8.3%, *p* < 0.001) and *BRCA1/2* variants (14.1% vs. 5.4%, *p* < 0.001) compared to the patients with a low expression (Ki-67 index ≤ 20%.) of Ki-67 (*n* = 374).

#### 2.3.4. Association between Deleterious Germline Variants and Hereditary Risk

Of all patients included in this study, 56.6% (396/700) were identified as hereditary high-risk, and 43.4% (304/700) were hereditary low-risk. The selection criteria for hereditary high-risk patients are presented in Table 2. Patients at hereditary high risk showed higher variant rates of panel-gene (18.2% vs. 6.6%, *p* < 0.001) and *BRCA1/2* variants (14.1% vs. 3.3%, *p* < 0.001; Figure 6a). We used the following clinical features associated with hereditary high risk to stratify BC patients into seven combined-risk groups: Early-onset of breast cancer, positive family history of any cancer, and TNBC. The variant rates of panel-gene (*p* = 0.001, Figure 6b) and *BRCA1/2* (*p* < 0.001, Figure 6c) increased with the number of features presented. For the patients presenting all three features (*n* = 13), the variant rates of panel-gene and *BRCA1/2* reached 53.8% (7/13).

The distribution of deleterious variants according to hereditary risk is shown in Figure 6d. A total of 74 variants were detected in patients at hereditary high risk, and mainly included *BRCA1* (39.2%, 29/74), *BRCA2* (36.5%, 27/74), *PALB2* (8.1%, 6/74), and *TP53* (4.1%, 3/74). There were 22 variants in patients at hereditary low-risk, and the common variants in these patients included *BRCA2* (45.5%, 10/22), *NBN* (13.6%, 3/22), and *PALB2* (9.1%, 2/22). *BRCA1* and *TP53* variants were all detected in patients at hereditary high risk (*BRCA1*: 7.3%; *TP53*: 0.8%). *BRCA2* variants were more prevalent in patients at hereditary high risk (6.8% vs. 3.3%, *p* = 0.039). *PALB2* variants were enriched in patients at hereditary high risk (1.5% vs. 0.7%, *p* = 0.477).

The clinicopathological predictors of deleterious germline variants in panel-gene, HRR-related gene, and *BRCA1/2* were consistent, including younger age at initial diagnosis of BC, positive family history of any cancer, TN phenotype, high expression of Ki-67, and hereditary high risk. The Association between germline variants and clinicopathological factors in overall recruited patients is shown in Table 3.

## 3. Discussion

In this study, we observed that 13.1% of Chinese BC patients carried deleterious germline variants within a large clinic-based cohort by the NGS in 32 cancer-related genes, which was roughly consistent with the study conducted by Chen et al. [16]. Sun et al. also conducted a multicenter study using 62-gene panel NGS among consecutive unselected 8085 Chinese BC patients and found that the pathogenic germline variant rate was 9.2% which was slightly different from our result [3]. It might cause by more patients with TN phenotype in our study than Sun et al. (23.4% vs. 13.7%). According to previous studies, TNBC patients presented a higher rate of germline variants than those with other molecular subtypes [5,17]. Germline *BRCA1/2* variants were observed in 9.4% of patients in our study: 5.3% in *BRCA2* and 4.1% in *BRCA1*. Because 56.6% (396/700) of the patients included in our study were at hereditary high risk, the variant rate of *BRCA1/2* was higher than that in the previous studies conducted among unselected patients [2,3,5,16,18,19]. The prevalence of *BRCA1/2* variants was affected by ethnic and geographical factors. The German Consortium for Hereditary Breast and Ovarian Cancer (GC-HBOC) conducted a comprehensive *BRCA* screening and revealed a higher mutation frequency in *BRCA1* than that in *BRCA2* [20]. However, Santonocito et al. reported that *BRCA2* variants were more commonly detected than *BRCA1* variants based on a large cohort of patients coming from Central-South Italy [21]. Hall et al. also found that germline *BRCA2* variants were more likely to be screened among the Chinese population than in Caucasian, particularly Latin American, patients [22]. This suggested that underlying specific *BRCA2* variant sites were abundant in the Chinese population. As a result, 14 (14/35, 40%) novel deleterious variants in *BRCA2* were firstly observed in our study among Chinese BC patients.

Current evidence demonstrated deleterious germline *BRCA1/2* variants were effective therapeutic genetic targets in Her2-negative early and metastatic BC patients [7,8]. Considering cost-effectiveness, it was rational to select a high-risk group of patients to undergo cancer-related gene screening via the NGS procedure in clinical practice. This study showed that *BRCA1/2* variants carriers could be frequently identified among BC patients with certain predictive factors. However, the characteristics of the carriers of *BRCA1* and *BRCA2* variants exhibited some unique features. *BRCA1* variant carriers were more likely to be early onset of BC, with a positive family history of any cancer and TN phenotype. *BRCA2* variant carriers showed a higher proportion of HR-positive phenotype. Both *BRCA1* and *BRCA2* variants were observed in a few patients with Her2-positive BC. Furthermore, among the patients presenting all three features associated with hereditary high risk, the variant rate of *BRCA1/2* even reached over 50%. This was comparable to the data from the previous studies conducted on Chinese patients with hereditary high risk BC [4,23,24,25]. Additionally, published studies observed that bilateral BC and personal history of secondary cancer were predictive factors for *BRCA1/2* variants, especially in patients at hereditary high risk [18,19,25]. However, only 3.9% (27/700) of the patients included in this study were bilateral, and 3.6% (25/700) had a personal history of secondary malignant tumors. For this reason, we did not observe the association between laterality of BC as well as personal cancer history and deleterious germline variants, including *BRCA1/2*. Consistent with Lang et al. [18], we found that *BRCA1/2* variants were much more commonly detected in BC patients presenting high levels of Ki-67 index (69.7% vs. 30.3%). Based on these findings, comprehensive predictive factors for *BRCA1/2* variants could be further understood so as to define consistent clinical criteria for decision-making to undergo genetic counseling and testing for high-risk BC patients.

Among non-*BRCA1/2* genes, the variants in *PALB2* were mostly detected with a rate of 1.1% (8/700). The relatively high prevalence of *PALB2* was consistent with the results from the previous study conducted in China [3] and the West [5] among unselected patients with BC. Of BC patients carrying *PALB2* variants, 75% (6/8) were HR-positive which was consistent with the data reported by Antoniou et al. among BC patients from Poland [26]. The deleterious germline variants in *PALB2* are an important cause of hereditary BC [26,27]. We also found that 75.0% (6/8) of *PALB2* carriers were at hereditary high risk. Contralateral BC was reported in 10% of *PALB2* variants carriers in the previous study [26]. However, contralateral BC was not observed in all eight *PALB2* carriers from our study. 

*NBN*, as a gene involved in DNA double-strand break repair, is responsible for encoding the protein nibrin. The founder *NBN* variants were detected more frequently in Polish and Finnish BC cases [28], and only c.657del5 was known as a pathogenic variant and associated with BC risk [29]. Given the limited evidence, the NCCN guidelines do not recommend breast cancer risk management for carriers of an *NBN* variant beyond c.657del5 [10]. Interestingly, we observed a recurrent *NBN* missense variant (c.1912T > C (p.S638P)) in four (4/4100.0%) Chinese BC patients: Three were HR-positive BC and one was TNBC. A functional test conducted by Wang et al. showed that this variant might impair the function of the NBN complex and thus affect DNA damage repair [30]. Therefore, we speculated that *NBN* c.1912T > C (p.S638P) was a deleterious germline variant specific to the Chinese population, especially HR-positive BC patients. However, further investigations are required to observe the association between this variant and BC risk. 

The variants in *TP53* were detected in 3 out of 700 (0.43%) BC patients in this study. Li-Fraumeni syndrome is a highly penetrant cancer syndrome associated with deleterious *TP53* variants [31]. According to the NCCN guidelines [10], none of the *TP53* variant carriers in this study met the classic criteria of Li-Fraumeni syndrome, while all of them met the Chompret criteria of BC before 31 years of age. In this study, 66.7% (2/3) of *TP53* variant carriers presented a Her2-positive phenotype. The amplification of Her2 may arise in conjunction with germline *TP53* variants. Hu et al. conducted a case-control study based on a large clinical genetic testing cohort and found that *TP53* variants were enriched in Her2-positive BC (OR, 22.71, 95%CI, 10.45–45.49) [32]. Furthermore, *TP53* variants were only detected in patients at hereditary high risk. Li et al. also reported that *TP53* variants were the most common non-*BRCA1/2* variants in Chinese BC patients at hereditary high risk with a rate of 1.9% [4]. Therefore, multiple-gene sequencing and counseling involving *TP53* should be highly recommended for Her2-positive BC patients at hereditary high risk. For other high-penetrance genes, no deleterious variant was detected in *CDH1*, *PTEN*, and *STK11*.

The variants of other genes related to DNA repair were also detected, such as *MRE11A*, *BARD1*, *CHEK1*, *CHEK2*, *ATR*, *BRIP1*, *FANCL*, *RAD51*, and *RAD54L*. The protein-trunking *CHEK2* variant c.1100delC, which had an increased risk for BC [33], was not observed among the patients in our study. Consistent with the published study [32], *CHEK2* variants were enriched in ER-positive BC, and *BARD1* variants were enriched in TNBC. Additionally, we found that *BARD1* variants were only detected in patients older than 55 years which was different from *BRCA1*. This suggested the need for multiple-gene testing in elderly patients with TNBC. *RAD51* plays an important role in DNA repair by HRR. In our study, *RAD51* variants were detected in two (0.3%, 2/700) BC patients: One with TNBC, and the other with HR-positive/Her2-negative BC. Ma et al. found that even about 2.5% of Chinese TNBC patients carried *RAD51D* pathogenic variants, which surpassed the rates for Caucasian and African American TNBC patients [34]. Some research to explore the impact on the clinical and molecular characteristics of these variants is urgently needed.

There are some limitations to this study. First, a part of cancer-susceptibility genes is not included in the multiple-gene panel, such as *NF1* and some of the mismatch repair genes (*MLH1*, *MSH2*, *MSH6*, *PMS2*, and *EPCAM*). Second, the clinicopathological characteristics of non-*BRCA1/2* variants remain unclear because of the low frequency of deleterious germline variants of these genes. In the following studies, we will continue to expand the sample size to select the variants with clinical utility and provide evidence for clinical decision-making.

## 4. Materials and Methods

### 4.1. Patients and Clinicopathological Factors

Between November 2020–February 2022, patients diagnosed with histology-confirmed BC in the Breast Disease Center of Peking University People’s Hospital were consecutively included. Patients without formal informed consent were excluded.

Clinicopathological factors of patients were extracted from HIS (Hospital Information System), including gender, age at initial diagnosis of BC, family history of any cancer (history of malignant tumor in the first-, second-, or third-degree blood relatives of the patients), personal history of cancer, laterality of BC, histology type and grade according to criteria of WHO (World Health Organization) [35] and TNM (tumor-nodal-metastasis) stage according to the AJCC (American Joint Committee on Cancer) Staging Manual: 8th edition [36]. The expression status of the ER, progesterone receptor (PR), and Her2 were interpreted according to ASCO/CAP (American Society of Clinical Oncology/College of American Pathologists) [37,38]. ER-positive and/or PR-positive were categorized as HR-positive, otherwise HR-negative. Then the molecular types of breast cancer were divided into the following four categories according to the status of HR and Her2: HR+Her2−, HR+Her2+, HR−Her2+, and HR−Her2−.

### 4.2. Hereditary High Risk Assessment

Based on NCCN guidelines for genetic/familial high-risk assessment on breast, ovarian, and pancreatic cancer [10], participants who met one or more of the following criteria were considered to be at hereditary high risk: (1) Diagnosed with BC at age ≤ 45 years, (2) diagnosed with BC at age 46–50 year with second breast cancer diagnosed at any age or ≥1 close blood relative with *BRCA*-related cancer (BC, epithelial ovarian cancer, including fallopian tube cancer or peritoneal cancer, exocrine pancreatic cancer, and prostate cancer.) at any age, (3) ≥1 close blood relative diagnosed with *BRCA*-related cancer at age ≤ 50 years, (4) ≥2 non-close blood relatives diagnosed with *BRCA*-related cancer at any age, (5) diagnosed at age ≤ 60 years with triple-negative BC, (6) diagnosed at any age with male BC, (7) diagnosed at any age with *BRCA*-related cancer.

### 4.3. DNA Extraction

After signing a formal informed consent form, 5 mL of fresh peripheral venous whole blood was collected from each patient and transferred into an ethylenediaminetetraacetic acid (EDTA) tube at 4 °C. The blood samples were sent to the Clinical Laboratory of Peking University People’s Hospital on the same day. Next, DNA samples were extracted from the peripheral blood by QIAamp DNA Mini Kit (Qiagen, Dusseldorf, Germany). Finally, the concentration and purity of the DNA samples were tested using a NanoDrop spectrophotometer (Thermo Fisher Scientific, Wilmington, DE, USA). The amplicon libraries were established by Ion AmpliSeq™ Library Kit 2.0 and Ion AmpliSeq™ custom primer pools (Analyses Technology Co., Ltd. Tokyo, Japan).

### 4.4. NGS Assay and Variant Classification

A multiple-gene panel consisted of 32 cancer-related genes, including 24 cancer-susceptibility genes and 19 HRR-related genes (shown in Appendix A). Based on Ion Torrent S5™ platform (Thermo Fisher Scientific, Wilmington, DE, USA), gene testing was performed. After trimming adapter sequences and removing polyclonals, test fragments, and low-quality reads, the sequences were aligned to *human* genome reference version hg19/GRCh37. 

The variants were filtered using the current databases, including 1000 Genomes (https://www.internationalgenome.org/data/, accessed on 10 December 2020), dbSNP142 (National Center for Biotechnology Information, http://www.ncbi.nlm.nih.gov/SNP/, accessed on 10 December 2020), NHLBI Grand Opportunity Exome Sequencing Project (ESP6500) (https://esp.gs.washington.edu/drupal/, accessed on 10 December 2020), Pfam (http://pfam.xfam.org, accessed on 10 December 2020), ExAC03 (http://exac.broadinstitute.org, accessed on 10 December 2020), Uniprot (http://www.uniprot.org, accessed on 10 December 2020), and Online Mendelian Inheritance in Man (OMIM) (https://omim.org/statistics/update, accessed on 10 December 2020). 

Then, evidence of pathogenicity of filtered variants was collected by different databases and predictive software. The conservation analysis of identified variants was performed by using phyloP46way_placental [39] and likelihood ratio tests (LRT) [40]. Pathogenicity prediction was made by the following software: SIFT (http://sift.jvvi.org/, accessed on 15 December 2020) [41], PolyPhen2 (http://genetics.bwh.harvard.edu/pph2/, accessed on 15 December 2020) [42] and Mutation Taster (http://www.mutationtaster.org/, accessed on 15 December 2020). Variants were referenced to the gene-specific mutation databases and published studies by Clinvar (https://www.ncbi.nlm.nih.gov/clinvar/, accessed on 15 December 2020), BRCA Exchange (http://brcaexchange.org, accessed on 15 December 2020), and Human Gene Mutation Database (HGMD) (http://www.hgmd.org/, accessed on 15 December 2020). 

Finally, the variants were interpreted for their pathogenicity according to the summary of the evidence, American College of Medical Genetics and Genomics (ACMG) recommendations [43], and Chinese expert consensus [44]. The pathogenicity was classified as pathogenic, likely pathogenic, uncertain significance, likely benign, and benign. Only pathogenic and likely pathogenic variants, both of which were classified as deleterious variants, were further analyzed in this study. 

### 4.5. Statistical Analysis

Continuous variables were reported as mean and standard deviation, whereas categorical variables were reported as percentages. Statistical differences in the distribution of continuous and categorical variables were conducted by T-test and chi-square test, respectively. Two-tailed *p*-values < 0.05 were considered statistically significant. All analyses were conducted using R software (Beijing China, http://www.Rproject.org, accessed on 15 May 2022).

## 5. Conclusions

We further expand the spectrum of germline deleterious variants in Chinese BC patients using a large clinic-based cohort by the NGS. The clinicopathological predictors of variants were identified to facilitate clinical genetic testing and counseling for appropriate individuals. However, some cancer-related genes were not included in the sequencing panel. Future studies with larger multiple-gene panels by NGS or third-generation sequencing are urgently needed to continue to expand the germline variant spectrum of BC patients in China.

## Figures and Tables

**Figure 1 ijms-23-11266-f001:**
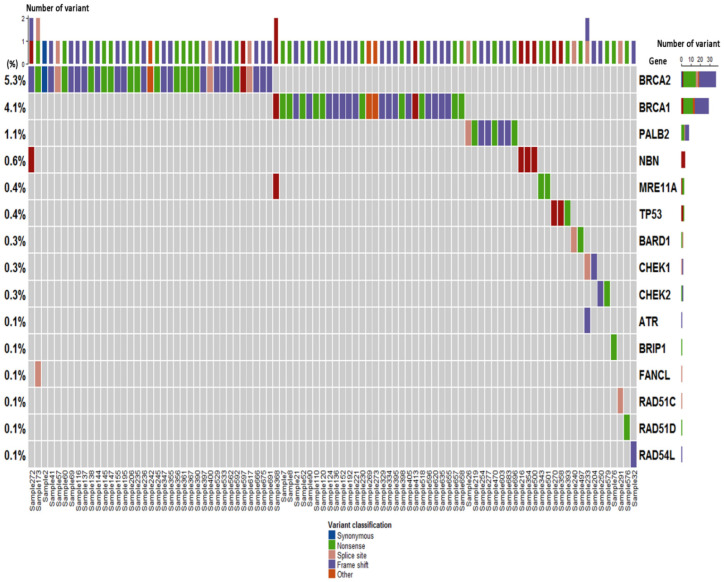
The spectrum of deleterious germline variants in 32 cancer-related genes among 700 Chinese breast cancer patients.

**Figure 2 ijms-23-11266-f002:**
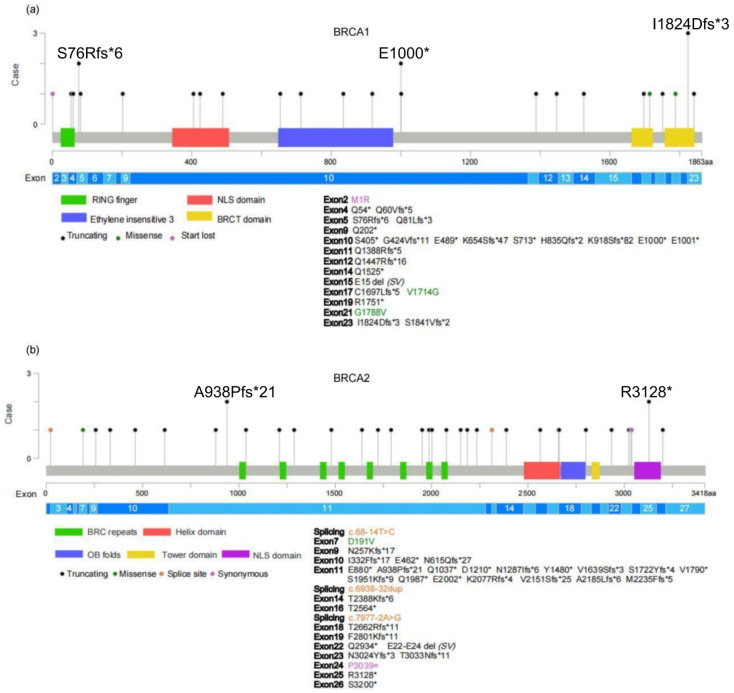
Locations of variants in the BRCA1/2 proteins. (**a**) The variants in BRCA1 protein. E15 deletion (*n* = 1) is not presented in the graph. (**b**) The variants in BRCA2 protein. E22-E24 deletion (*n* = 1) is not presented in the graph. Abbreviation: BRCT = the BRCA carboxyl terminus domain; NLS = nuclear localization signal; OB folds = oligonucleotide binding-folds; SV = structure variants; * = nonsense variant.

**Figure 3 ijms-23-11266-f003:**
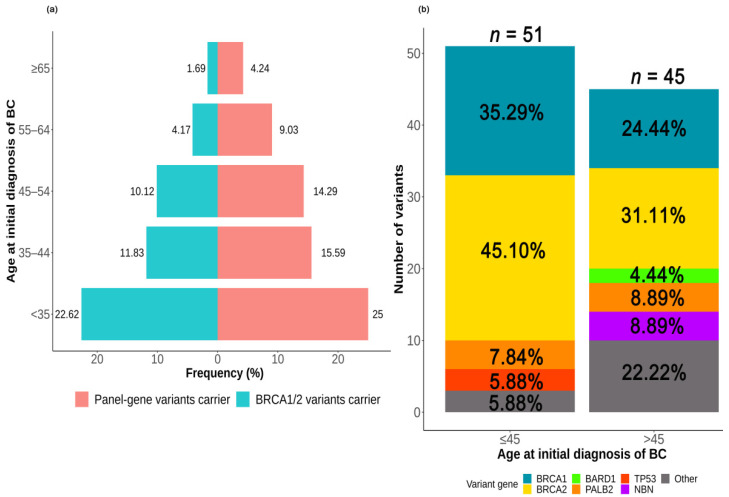
Association between deleterious germline variants and age at the initial diagnosis of breast cancer. (**a**) The variant rates according to age at initial diagnosis of breast cancer, (**b**) distribution of the variants according to age at initial diagnosis of breast cancer. Abbreviation: BC = breast cancer.

**Figure 4 ijms-23-11266-f004:**
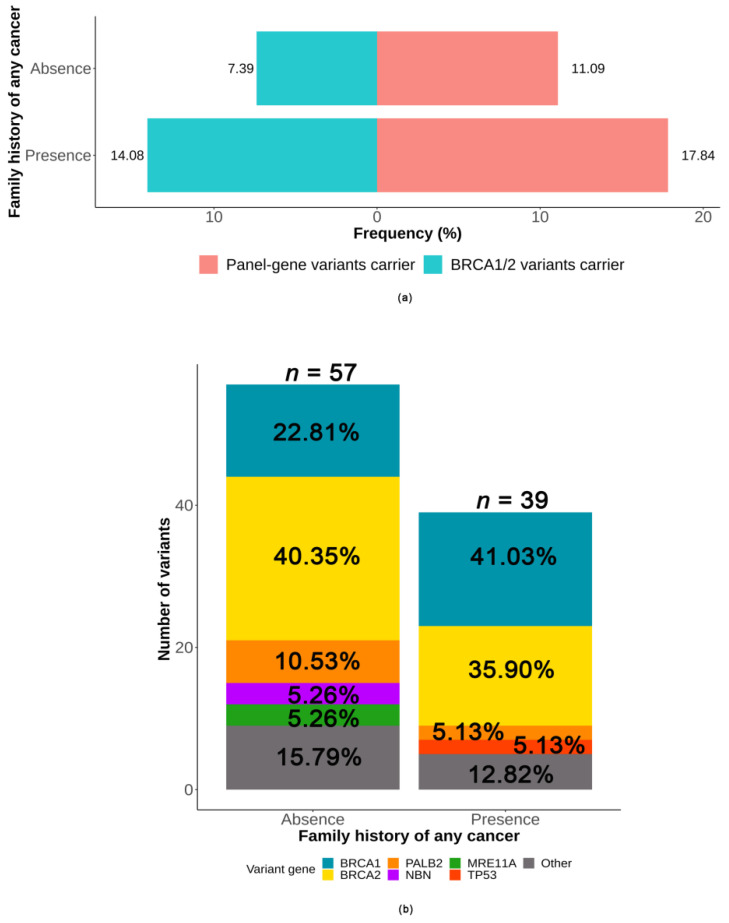
Association between deleterious germline variants and family history of any cancer. (**a**) The variant rates according to family history of any cancer, (**b**) distribution of the variants according to family history of any cancer.

**Figure 5 ijms-23-11266-f005:**
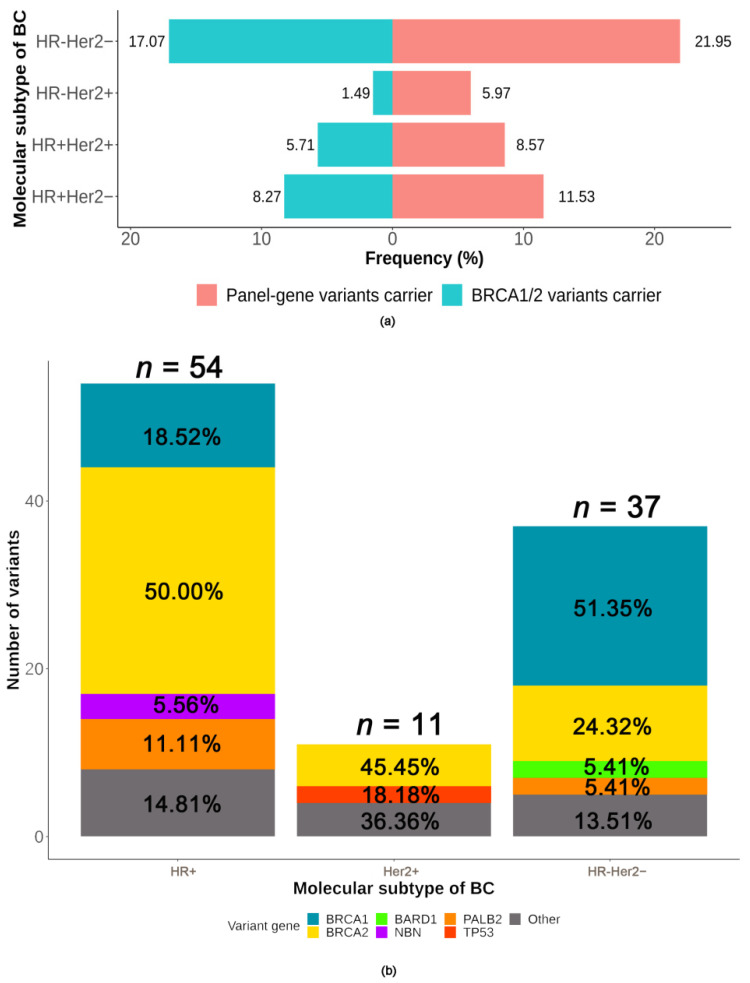
Association between deleterious germline variants and molecular subtypes of breast cancer. (**a**) The variant rates according to molecular subtypes of breast cancer, (**b**) distribution of the variants according to molecular subtypes of breast cancer. Abbreviations: BC = breast cancer; HR = hormone receptor; Her2 = human epidermal growth factor receptor.

**Figure 6 ijms-23-11266-f006:**
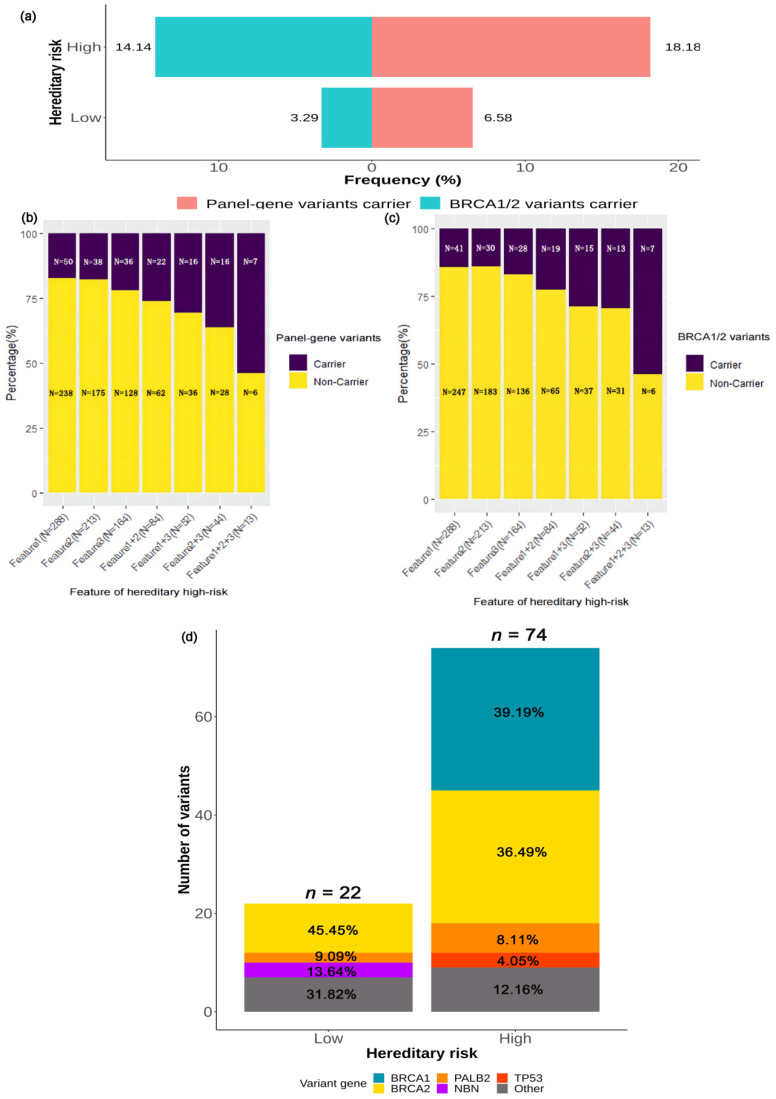
Association between deleterious germline variants and hereditary risk. (**a**) The variant rates according to hereditary risk. The variant rates of (**b**) panel-gene and (**c**) *BRCA1/2* according to specifically defined features of hereditary high risk. Feature 1: Early-onset (age at initial diagnosis of breast cancer ≤45 years), Feature 2: Positive family history of any cancer, Feature 3: Triple-negative breast cancer. (**d**) Distribution of the variants according to hereditary risk.

**Table 1 ijms-23-11266-t001:** Characteristics of 700 patients with BC included in the study.

Characteristics	No.	%
Gender		
Female	695	99.3
Male	5	0.7
Nationality		
Han	696	99.4
Mongolian	2	0.3
Uyghur	2	0.3
Age at the initial diagnosis of BC		
<35	84	12.0
35–44	186	26.6
45–54	168	24.0
55–64	144	20.6
≥65	118	16.9
Family history ^1^		
No	487	69.6
Yes	213	30.4
Personal history of cancer ^2^		
No	675	96.4
Yes	25	3.6
Hereditary risk ^3^		
Low	304	43.4
High	396	56.6
Laterality of BC		
Unilateral	673	96.1
Bilateral	27	3.9
Histology type		
DCIS	59	8.4
Ductal	534	76.3
Lobular	19	2.7
Mixed	57	8.1
Other ^4^	31	4.4
Histology grade		
I	45	6.4
II	390	55.7
III	185	26.4
UNK	80	11.4
Tumor stage		
Tis	59	8.4
T1	327	46.7
T2	266	38.0
T3–T4	48	6.9
Nodal status		
Negative	419	59.9
Positive	281	40.1
TNM stage		
0	59	8.4
I	222	31.7
II	301	43.0
III	93	13.3
IV	25	3.6
Molecular subtype		
HR+Her2−	399	57.0
HR+Her2+	70	10.0
HR−Her2+	67	9.6
HR−Her2−	164	23.4
Ki-67 Index		
≤20%	374	53.4
>20%	326	46.6

^1^ Family history: history of any cancer in the first-, second-, or third-degree blood relatives of the patients. ^2^ Personal history of cancer: Personal history of primary cancer except for breast cancer. ^3^ Hereditary risk: Hereditary risk was assessed based on the NCCN guidelines for genetic/familial high-risk assessment on breast, ovarian, and pancreatic cancer. ^4^ Other: Special histology types of invasive breast cancer except for ductal and lobular carcinoma. Abbreviation: BC = breast cancer; DCIS = ductal carcinoma in situ; Her2 = human epidermal growth factor receptor 2; HR = hormone receptor; NCCN = National Comprehensive Cancer Network; No. = number of patients; TNM = tumor-lymph node-metastasis; UNK = unknown.

**Table 2 ijms-23-11266-t002:** Selection criteria and numbers for the patients at hereditary high risk.

Criteria of Hereditary High Risk		No. (%)
C.1	Diagnosed with breast cancer at age ≤ 45 years.	288 (41.1)
C.2	Diagnosed with breast cancer at age 46–50 years with one of the following:	12 (1.7)
	(1) A second breast cancer diagnosed at any age;	
	(2) ≥1 close blood relative with *BRCA*-related cancer ^1^ at any age.	
C.3	≥1 close blood relative diagnosed with *BRCA*-related cancer ^1^ at age ≤ 50 years.	54 (7.7)
C.4	≥2 Non-close blood relatives diagnosed with *BRCA*-related ^1^ cancer at any age.	9 (1.3)
C.5	Diagnosed at age ≤ 60 years with triple-negative breast cancer.	120 (17.1)
C.6	Diagnosed at any age with male breast cancer.	5 (0.7)
C.7	Diagnosed at any age with *BRCA*-related cancer ^1^.	5 (0.7)

^1^*BRCA*-related cancer: Breast cancer, epithelial ovarian cancer (including fallopian tube cancer or peritoneal cancer), exocrine pancreatic cancer, and prostate cancer.

**Table 3 ijms-23-11266-t003:** The association between deleterious germline variants and clinicopathological characteristics in 700 Chinese patients with BC included in the study.

Characteristics	No.	Panel-Gene Variants			HRR-Gene Variants			*BRCA1/2* Variants		
		Carrier *n* = 92	Non-Carrier *n* = 608	*p*	Carrier *n* = 89	Non-Carrier *n* = 611	*p*	Carrier *n* = 66	Non-Carrier *n* = 634	*p*
Gender (%)										
Female	695	91 (98.9)	604 (99.3)	1.000	88 (98.9)	607 (99.3)	1.000	66 (100.0)	629 (99.2)	1.000
Male	5	1 (1.1)	4 (0.7)		1 (1.1)	4 (0.7)		0 (0.0)	5 (0.8)	
Nationality (%)										
Han	696	92 (100.0)	604 (99.3)	NA	89 (100.0)	607 (99.3)	NA	66 (100.0)	630 (99.4)	NA
Mongolian	2	0 (0.0)	2 (0.3)		0 (0.0)	2 (0.3)		0 (0.0)	2 (0.3)	
Uyghur	2	0 (0.0)	2 (0.3)		0 (0.0)	2 (0.3)		0 (0.0)	2(0.3)	
Age (%)										
<35	84	21 (22.8)	63 (10.4)	<0.001	21 (23.6)	63 (10.3)	<0.001	19 (28.8)	65 (10.3)	<0.001
35–44	186	29 (31.5)	157 (25.8)		26 (29.2)	160 (26.2)		22 (33.3)	164 (25.9)	
45–54	168	24 (26.1)	144 (23.7)		24 (27.0)	144 (23.6)		17 (25.8)	151 (23.8)	
55–64	144	13 (14.1)	131 (21.5)		13 (14.6)	131 (21.4)		6 (9.1)	138 (21.8)	
≥65	118	5 (5.4)	113 (18.6)		5 (5.6)	113 (18.5)		2 (3.0)	116 (18.3)	
Family history ^1^ (%)										
No	487	54 (58.7)	433 (71.2)	0.021	53 (59.6)	434 (71.0)	0.038	36 (54.5)	451 (71.1)	0.008
Yes	213	38 (41.3)	175 (28.8)		36 (40.4)	177 (29.0)		30 (45.5)	183 (28.9)	
Personal history of cancer ^2^ (%)										
No	675	88 (95.7)	587 (96.5)	0.897	86 (96.6)	589 (96.4)	1.000	64 (97.0)	611 (96.4)	1.000
Yes	25	4 (4.3)	21 (3.5)		3 (3.4)	22 (3.6)		2 (3.0)	23 (3.6)	
Hereditary risk ^3^ (%)										
Low	304	20 (21.7)	284 (46.7)	<0.001	20 (22.5)	284 (46.5)	<0.001	10 (15.2)	294 (46.4)	<0.001
High	396	72 (78.3)	324 (53.3)		69 (77.5)	327 (53.5)		56 (84.8)	340 (53.6)	
Laterality (%)										
Unilateral	673	88 (95.7)	585 (96.2)	0.896	86 (96.6)	587 (96.1)	0.899	64 (97.0)	609 (96.1)	0.877
Bilateral	27	4 (4.3)	23 (3.8)		3 (3.4)	24 (3.9)		2 (3.0)	25 (3.9)	
Histology type (%)										
DCIS	59	4 (4.3)	55 (9.0)	0.589	4 (4.5)	55 (9.0)	0.637	4 (6.1)	55 (8.7)	0.773
Ductal	534	74 (80.4)	460 (75.7)		71 (79.8)	463 (75.8)		51 (77.3)	483 (76.2)	
Lobular	19	3 (3.3)	16 (2.6)		3 (3.4)	16 (2.6)		3 (4.5)	16 (2.5)	
Mixed	57	8 (8.7)	49 (8.1)		8 (9.0)	49 (8.0)		6 (9.1)	51 (8.0)	
Other ^4^	31	3 (3.3)	28 (4.6)		3 (3.4)	28 (4.6)		2 (3.0)	29 (4.6)	
Histology grade (%)										
I	45	2 (2.2)	43 (7.1)	0.115	2 (2.2)	43 (7.0)	0.144	1 (1.5)	44 (6.9)	0.131
II	390	53 (57.6)	337 (55.4)		51 (57.3)	339 (55.5)		37 (56.1)	353 (55.7)	
III	185	30 (32.6)	155 (25.5)		29 (32.6)	156 (25.5)		23 (34.8)	162 (25.6)	
UNK	80	7 (7.6)	73 (12.0)		7 (7.9)	73 (11.9)		5 (7.6)	75 (11.8)	
Tumor stage (%)										
Tis	59	4 (4.3)	55 (9.0)	0.332	4 (4.5)	55 (9.0)	0.415	4 (6.1)	55 (8.7)	0.454
T1	327	41 (44.6)	286 (47.0)		40 (44.9)	287 (47.0)		27 (40.9)	300 (47.3)	
T2	266	41 (44.6)	225 (37.0)		39 (43.8)	227 (37.2)		31 (47.0)	235 (37.1)	
T3-T4	48	6 (6.5)	42 (6.9)		6 (6.7)	42 (6.9)		4 (6.1)	44 (6.9)	
Nodal status (%)										
Negative	419	47 (51.1)	372 (61.2)	0.084	46 (51.7)	373 (61.0)	0.092	33 (50.0)	386 (60.9)	0.133
Positive	281	45 (48.9)	236 (38.8)		43 (48.3)	238 (39.0)		33 (50.0)	248 (39.1)	
TNM stage (%)										
0	59	4 (4.3)	55 (9.0)	0.073	4 (4.5)	55 (9.0)	0.098	4 (6.1)	55 (8.7)	0.092
I	222	25 (27.2)	197 (32.4)		24 (27.0)	198 (32.4)		15 (22.7)	207 (32.6)	
II	301	51 (55.4)	250 (41.1)		49 (55.1)	252 (41.2)		39 (59.1)	262 (41.3)	
III	93	11 (12.0)	82 (13.5)		11 (12.4)	82 (13.4)		7 (10.6)	86 (13.6)	
IV	25	1 (1.1)	24 (3.9)		1 (1.1)	24 (3.9)		1 (1.5)	24 (3.8)	
Subtype (%)										
HR+Her2−	399	46 (50.0)	353 (58.1)	0.001	46 (51.7)	353 (57.8)	<0.001	33 (50.0)	366 (57.7)	<0.001
HR+Her2+	70	6 (6.5)	64 (10.5)		6 (6.7)	64 (10.5)		4 (6.1)	66 (10.4)	
HR−Her2+	67	4 (4.3)	63 (10.4)		2 (2.2)	65 (10.6)		1 (1.5)	66 (10.4)	
HR−Her2−	164	36 (39.1)	128 (21.1)		35 (39.3)	129 (21.1)		28 (42.4)	136 (21.5)	
Ki-67 Index (%)										
≤20%	374	31 (33.7)	343 (56.4)	<0.001	31 (34.8)	343 (56.1)	<0.001	20 (30.3)	354 (55.8)	<0.001
>20%	326	61 (66.3)	265 (43.6)		58 (65.2)	268 (43.9)		46 (69.7)	280 (44.2)	

^1^ Family history: History of any cancer in the first-, second-, or third-degree blood relatives of patients. ^2^ Personal history of cancer: Personal history of primary cancer except for breast cancer. ^3^ Hereditary risk: Hereditary risk was assessed based on the NCCN guidelines for genetic/familial high-risk assessment on breast, ovarian, and pancreatic cancer. ^4^ Other: Special histology type of invasive breast cancer except for ductal and lobular carcinoma. Abbreviation: BC = breast cancer; DCIS = ductal carcinoma in situ; Her2 = human epidermal growth factor receptor 2; HR = hormone receptor; HRR = homologous recombination repair; NA = not applicable; NCCN = National Comprehensive Cancer Network; TNM = tumor-lymph node-metastasis; UNK = unknown.

## Data Availability

All data generated or analyzed during the study are included in the published paper.

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
