# Peer review of "Germline Variants in 32 Cancer-Related Genes among 700 Chinese Breast Cancer Patients by Next-Generation Sequencing: A Clinic-Based, Observational Study"

_ijms, 2022, doi:10.3390/ijms231911266_

Round 1

Author Response

Thank you very much for taking the time to review this manuscript. We appreciate your comments and suggestions. Your constructive suggestios help us better improve the manuscript. We have tried to think and response carefully to your comments point-by-point (detailed in attached PDF file). We are very glad to communicate with you, and thanks again for your great contribution to our work.

Reviewer 2 Report

The study by Liu Yang et al investigates the prevalence of germline variants in a Chinese population of 700 breast cancer patients. Their goal was to expand the spectrum of deleterious germline variants in Chinese BC patients. The content of the paper as high potential but needs some revisions.

Minor Revision

1. Improve the English

2. They found a higher rate of the variant in the BRCA2 gene compared to BRCA1. This is not the first time. Example Santonocito, C. et al had similar results in a population of South Italy breast cancer patients. Need to check other reference like this one to add in the discussion (Spectrum of Germline BRCA1 and BRCA2 Variants Identified in 2351 Ovarian and Breast Cancer Patients Referring to a Reference Cancer Hospital of Rome. Cancers 2020, 12, 128)

Major Revision

1. This would be a major revision only if among the 700 Chinese patients analyzed in the paper there is more than one ethnic group.It is crucial to know and understand the diversity of a population when you are testing the prevalence of variants associated with a specific disease. If more than one ethnic group is present, the data have to be analyzed taking that into account ad they have done already considering gender and age.

Author Response

Thank you for taking the time to read this manuscript. We are very excited by your recognition of our work. We have carefully read your review comments and tried to make prudent point-by-point responses (detailed in the atteched PDF). We really think your comments are very constructive and provide a direction for our following research. We have revised the manuscript according to your comments. Thank you again for your great contribution to our work.

Round 2

Reviewer 1 Report

The authors improved the manuscript very much and addressed the majority of the concerns presented.